# Dynamics of Biochemical Parameters, Inflammatory and Stress Markers in Lambs Undergoing Caudectomy Using Two Different Methods

**DOI:** 10.3390/ani15172614

**Published:** 2025-09-06

**Authors:** Giovannantonio Pilo, Giuseppe Tedde, Angelo Peli, Pier Attilio Accorsi, Gavina Carta, Loredana Secchi, Giulia Franzoni, Paola Nicolussi

**Affiliations:** 1Department of Animal Welfare and Clinic Diagnostic, Istituto Zooprofilattico Sperimentale della Sardegna, 07100 Sassari, Italy; gantonio.pilo@izs-sardegna.it (G.P.); giuseppe.tedde@izs-sardegna.it (G.T.); gavina.carta@izs-sardegna.it (G.C.); loredana.secchi@izs-sardegna.it (L.S.); paola.nicolussi@izs-sardegna.it (P.N.); 2Department for Life Quality Studies, Alma Mater Studiorum University of Bologna, 40126 Bologna, Italy; angelo.peli@unibo.it; 3Department of Veterinary Sciences, University of Bologna, 40126 Bologna, Italy; pierattilio.accorsi@unibo.it; 4Department of Animal Health, Istituto Zooprofilattico Sperimentale della Sardegna, 07100 Sassari, Italy

**Keywords:** sheep, caudectomy, inflammatory status, cortisol, biochemical profile

## Abstract

Many breeders have the habit of docking dairy sheep tails based on several reasons depending on the geographical area. Previous works link this practice with acute and long-term consequences, such as pain, onset of traumatic neuroma, and the higher incidence of rectal prolapses. We, therefore, carried out a case–control study to evaluate the impact of tail docking on stress markers, inflammatory markers, and biochemical parameters in lambs that have undergone caudectomy with two different practices (rubber rings or surgical practice), alongside controls. The results showed higher stress levels (wool cortisol values) and general inflammatory status in animals that underwent tail docking using rubber rings compared to controls. This study highlighted the detrimental consequences of this practice, the importance of a better evaluation of the “damage/benefit” of caudectomy in sheep, and the differences between the protocols used.

## 1. Introduction

Zootechnical practices such as tail docking are still in use in dairy sheep farming and can have repercussions on animal welfare, causing both acute and chronic pain [1,2,3]. Tail docking in lambs is practiced based on several reasons, depending on the geographical area. In Mediterranean Europe, where sheep are primarily raised for the production of milk, the tail is cut to enhance udder health, milking hygiene, and for greater ease during mating, although with controversial results in the literature [4,5,6].

In Northern Europe, where sheep are mainly raised for meat production, caudectomy is carried out in long-tailed breeds to prevent or minimise the onset of skin myiasis [3]. Caudectomy is widely considered effective in order to reduce the soiling of faeces on the fleece and, consequently, reduce the incidence of myiasis and promote greater udder hygiene [3,7]. Nevertheless, some authors have shown no correlation between tail length and fleece cleanliness, and other researchers have reported a higher incidence of myiasis in animals with very short tails, especially in cases of diarrhea [3,7,8]. In some studies, the evidence of a correlation between caudectomy and the reduction of myiasis is rather limited [3,9]. According to the Farm Animal Welfare Council (FAWC), caudectomy is often practiced more by tradition rather than a real need, and it is only partially effective in the control of myiasis [10].

This practice is associated not only with acute pain but also long-term consequences, such as traumatic neuroma and rectal prolapses [11,12].

Several methods are used to dock tails in sheep: surgical, docking iron, or applying constrictive rubber ring [3,9]. Directive 1998/58 EC does not regulate tail docking but states that the member states can adopt specific provisions concerning mutilations. The Italian law refers only to tail docking in bovine and not to other species, stating that this practice can only be performed for therapeutic purposes. Nevertheless, tail docking has been evaluated as mutilation by the ClassyFarm check list produced by the Italian National Centre of Reference for Animal Welfare (CReNBA) for assessing the welfare of dairy sheep [13]. Some authors suggest that tail docking should be performed only when there is real need (actual risk of myiasis), by properly trained personnel, and always with the use of anesthesia after the first week of life [9].

In Italy, the most used method for tail docking is the rubber ring, followed by the use of the Burdizzo pincer; other methods include the combination of the rubber ring with the Burdizzo pincer, thermocauterisation, termination with scissors or a knife, and more rarely, the surgical method. All these methods are usually performed during the first week of life, based on the belief that the lamb may be less sensitive to pain [14,15,16,17].

The degree of distress following tail docking in lambs is usually assessed by physiological parameters, especially circulating levels of cortisol, and behavioural observations [3,9,10,18,19]. Behavioural indicators of pain include lethargy, reduced movement, restlessness, rolling, kicking, loosening of the quarters, tail wagging, turning of the head, and changes in posture [3,14,16,17,18]. It was reported that the use of the rubber ring resulted in several behavioural changes indicative of acute pain [3,9]; nevertheless, the use of local anaesthetic resulted in reduced stress and fewer behavioural changes associated with this procedure [3,9]. So far, few studies have investigated in detail the effects of tail docking with diverse protocols on lambs’ health status and stress. Therefore, we carried out a control-group study to investigate the impact of caudectomy on stress and inflammatory markers, as well as on main biochemical parameters, comparing two different techniques: application of rubber ring and surgical amputation. Overall, we aimed to provide a better depiction of the impact of caudectomy on animal stress, inflammation, health status, and pain, to better evaluate the balance between the advantages and disadvantages of this procedure, by comparing two different techniques.

## 2. Materials and Methods

### 2.1. Ethical Statement and Study Design

The whole study was approved by the Ethics Committee of the Istituto Zooprofilattico Sperimentale (IZS) della Sardegna and authorised by the Italian Ministry of Health in accordance with Directive 26/2010/EU (implemented by Legislative Decree 26/2014) (authorization n. 472/2018-PR).

A case–control study was carried out at the experimental station of the IZS della Sardegna (‘Surigheddu’, Sassari, Italy), involving 21 lambs (<1 month old) randomly selected and equally allocated into three groups of 7 subjects each: A, control group (not treated); B, tail docking by rubber ring; and C, tail docking by surgical amputation. All the lambs were kept with their mothers and fed exclusively with maternal milk. Animals were housed in a room at the user’s establishment with comfortable bedding and 24 h water availability. Clinical visits and blood sample collections were performed before treatment (T_0_) and five times after caudectomy: T_1_ (36–40 h), T_2_ (3 days), T_3_ (7 days), T_4_ (14 days), and T_5_ (21 days). Whole blood was collected at T_0_, T_1_, T_2_, T_3_, T_4_, and T_5_, whereas wool for cortisol analysis was collected at T_0_ and T_4_. A schematic representation of the study design is presented in Figure 1. In addition, trained veterinarians monitored these animals for one week post-caudectomy using camera traps, in order to detect changes in behaviour indicating pain.

### 2.2. Cautectomy Procedures

In group B, lambs 4 days old were subjected to caudectomy through the application of the rubber ring, in the absence of anaesthetic. The ring was positioned between the fifth and sixth coccygeal vertebrae and left until the atrophied stump fell, usually after about 7–10 days.

The caudectomy by surgical amputation was instead performed on lambs 4–7 days old (group C). Caudectomy was performed between the fifth and sixth coccygeal vertebrae, after trichotomy and disinfection of the anatomical region concerned. Local anaesthesia was carried out by using 2% lidocaine (2 mL/head) subcutaneously, and after few minutes, surgery was performed. Once the intervertebral space was identified, through the use of surgical cutters, a compression of about two minutes was exerted in order to create haemostasis on the soft tissues and respective vessels. Exerting greater pressure on the surgical tool resulted in definitive amputation. Three simple surgical stitches with absorbable monofilament sutures were placed on the affected surface, in order to reduce the possibility of infections and promote the healing process. Topical antibiotic therapy was also practiced until complete recovery.

### 2.3. Blood Sampling

Individual blood sampling from the lambs were collected from the jugular vein, while an operator secured the animal by holding it in his arms, in order to reduce stress as much as possible. Disposable needles and vacutainer tubes were used for all animals. Blood samples were collected using serum vacutainers (Vacuette Tube, Greiner Bio-One, Kremsmünster, Austria) for biochemical profiling and electrophoretic analysis, whereas sodium citrate test tubes (Becton Dickinson, Milano, Italy) were used for fibrinogen analysis. Immediately after collection, blood samples were placed in a container with ice and transported to the laboratory within 2 h.

### 2.4. Biochemical and Electrophoretic Analysis

The serum was separated by laboratory centrifugation at 3500× *g* for 10 min at 4 °C, placed in 1.5 mL tubes, and stored at −20 °C until analysis. Biochemical parameters were analysed using a Dimension RXL chemistry analyser (Siemens, Munich, Germany), and serum protein electrophoresis was carried out using an INTERLAB G26 Automated Agarose Gel Electrophoresis Analyzer (Interlab, Rome, Italy), as previously described [20].

Fibrinogen was instead evaluated using an automated blood coagulation analyser (Sysmex CA-500, Kobe, Japan). The set of parameters used for biochemical and electrophoretic profiling are provided in Table 1.

### 2.5. Wool Cortisol Analysis

Wool was collected using an electric clipper, sampling at least 2–3 g of wool. The sampling area was identified in clean and dry areas of the mantle. Subsequent samples were taken in the same area in order to evaluate the cortisol concentration in the newly growing hair. Each sample collected was placed inside a plastic bag and identified with a progressive number and the collection date. All samples were sent to the Department of Veterinary Medical Sciences, University of Bologna (UNI EN/ISO 9001: 2015 certified [21]), to be processed.

The determination of the wool cortisol concentration was performed by radioimmunoassay techniques. In detail, wool samples were first prepared by manually cutting the material into 1–3 mm fragments and selecting 60 mg. Subsequently, they were incubated at +50 °C, for 18 h, under gentle agitation, after adding 5 mL of methanol. The samples were filtered to separate the hair from the methanol, and the latter was evaporated under an air flow suction hood. The dry residue was then dissolved in 1 mL of 0.05 M phosphate buffer, pH 7.5.

The determination of wool cortisol was carried out in duplicate: 100 μL of 3H-cortisol (30 pg/tube) and 100 μL of anti-cortisol antibody at a dilution of 1: 20,000 were added to 100 μL of phosphate buffer. After incubation at +4 °C for 18 h, the separation between bound and free hormone was performed by adding 1 mL of a 1% charcoal and 0.025% dextran solution to the samples, incubating at +4 °C to 15 min and centrifuging at 4000× *g* for 4 min at +4 °C. The supernatant, containing the hormone bound to the antibody, was poured into scintillation vials and used for the radioactivity count performed with a liquid scintillation β counter. The analysis validation parameters chosen are as follows: sensitivity 0.78 pg/vial, variability in the assay 6.8%, and between assays 9.3%. The conversion of the radioactivity of the samples (cpm/vial) into units pg/mg was performed using a specially designed calculation program (PRIAMO).

### 2.6. Statistical Analysis

Statistical analysis was conducted for each animal group at all sampling times. The Shapiro–Wilk test was used to evaluate the normal distribution of each independent variable. For biochemical and inflammatory markers, differences between values post-caudectomy (T_1_, T_2_, T_3,_ T_4,_ and T_5_) and values pre-treatment (T_0_) were investigated using the parametric ANOVA followed by Dunnett’s multiple comparison tests or the non-parametric Kruskal–Wallis test followed by Dunn’s multiple comparison test. For wool cortisol levels, differences between values post-caudectomy (T_4_) and values pre-treatment (T_0_) were investigated using an unpaired *t*-test. Differences were considered significant at *p* < 0.05. Data were graphically and statistically analysed with GraphPad Prism 10.01 (GraphPad Software Inc., La Jolla, CA, USA).

## 3. Results

In this work, the impact of two different protocols of caudectomy in lambs were investigated: rubber ring (in the absence of anaesthetic) and surgical treatment (with local anaesthetic).

As expected, both procedures resulted in changes in lambs’ behaviour indicating pain but with differences between groups. In detail, several animals belonging to group 3 (surgical treatment) presented lateral decubitus and reduced movement for 1–2 days post-treatment, whereas lambs that underwent tail docking by rubber rings presented lateral and sternal decubitus and reduced movement for almost 3 days post-treatment. Immediately after rubber ring application, we observed licking and scratching of the injured part, and subsequently abnormal gait (i.e., swaying, backward gait) and continuous change in posture (i.e., attempts to kick, trampling of the soil, and stretching).

In this study, the impact of these two procedures on animal health status and stress were thoroughly analysed, by monitoring the lamb’s changes in biochemical profile, serum electrophoresis, inflammatory markers, and wool cortisol levels at several times post-caudectomy.

First, the biochemical profile was monitored over time to assess tail docking impact on lambs’ health status. The impact of caudectomy on liver function and metabolic activity was investigated. Serum AST and ALP levels depict hepatocyte health and function, whereas serum bilirubin and cholesterol parameters are markers of liver’s metabolic capacity [22]. For all these parameters, no apparent abnormalities were observed outside normal physiological fluctuation (Figure 2). Then, the impact of tail docking on renal function was monitored by assessing the serum levels of urea and creatinine [23]. We observed that urea levels were unaffected in all groups during the study, with the exception of a modest decrease in urea levels at T_2_. Creatinine levels increased in all the groups, regardless of tail docking procedures; increased levels were observed at T_3_, T_4_, and T_5_ also in the control group (Figure 3). After that, the impact of an indicator of muscle injury was also investigated. Creatine phosphokinase (CPK) serum levels can be a useful indicator of muscle injury [24], and a modest rise in CPK serum levels was observed at 21 days (T_5_) in lambs that underwent caudectomy with rubber rings (group B), but not in the other groups (Figure 3). In addition, a slight decrease in glucose levels was observed in lambs that underwent tail docking with rubber rings, but not in the other groups (Figure 3).

Total protein levels and electrophoretic analysis were subsequently performed to evaluated changes in protein serum levels. Our analysis revealed only little fluctuation over time in group C (surgical treatment), with a transitory rise in β-globulin levels at T_2_ and T_4_, although without statistical significance. On the contrary, tail docking with rubber rings resulted in a higher modulation of serum protein values: at day 7 (T_3_), a significant drop in albumin levels was detected, concomitant with a rise in proportion of α2-globulins and β-globulins (Figure 4). Few changes were observed also in the control group, mainly due to a drop in γ-globulin levels at T_4_ and T_5_, likely due to decreased levels of γ-globulin in milk compared to colostrum (Figure 4).

Circulating levels of inflammatory markers (fibrinogen and transferrin) were also investigated. As presented in Figure 5, tail docking with rubber rings resulted in a higher percentage of transferrin at day 14 (T_4_), whereas no significant alteration was observed in the other groups. No significant modulation was detected in fibrinogen levels in any of the tested groups (Figure 5).

Then, wool cortisol levels were quantified. Wool samples were collected before caudectomy (T_0_) and 14 days after caudectomy (T_4_) to evaluate the levels of the stress marker cortisol. We observed that tail docking with rubber rings resulted in a significant rise in wool cortisol levels compared to pre-treatment levels, whereas no significant changes between time points were observed in the other groups (Figure 6).

## 4. Discussion

The tail is a very important anatomical part used by animals in ambulation, especially in running, to maintain balance and plays an important behavioural role in communication between animals. The tail is also a useful tool for removing insects and represents the attachment point of the muscles that regulate the correct functioning of the rectum. There is both behavioural and physiological evidence that tail docking is painful for sheep; both responses are reduced when pain relief is provided; thus, further research is required to justify tail docking of sheep as a routine practice [9].

In this work, we evaluated the impact of two methods of tail docking in sheep (rubber rings and surgical amputation). Several studies reported that both procedures resulted in animal pain [3,14,25], and accordingly, we observed changes in lambs’ behaviour indicative of pain in the first week post-caudectomy.

In this work, changes in biochemical parameters and stress markers were monitored over time, in order to provide a more detailed depiction of the impact of these procedures on animal pain, stress, and health status.

Alterations in kidney and liver functions were evaluated. The standard panel for renal function assessment included serum urea and creatinine levels [23], and we observed that urea values were unaffected in all groups during the study, with the exception of a modest and transitory decrease in group B (rubber ring) at T_2_. Creatinine levels rose in all groups, regardless of tail docking or the procedure used, and this likely reflects age-related changes. The rise in creatinine values might be due to poor creatinine clearance observed in the first few days of postnatal life or the inability of the neonatal kidney to completely excrete this molecule, which is partially reabsorbed by the immature tubule [25]. Overall, our data indicated that tail docking with either rubber rings or surgical treatments did not substantially impair renal filtration functions.

Four serum markers of liver functions were analysed: AST, ALT, bilirubin, and cholesterol. For all these parameters, no apparent abnormalities were observed outside normal physiological fluctuation, showing that tail docking with either rubber rings or surgical treatments did not substantially impair liver functions.

A rise in Creatine phosphokinase (CPK) serum levels was also quantified. This enzyme can be a useful indicator of muscle injury [24], and a modest but significant rise in CPK serum levels in lambs that have undergone caudectomy by rubber rings reflects the muscular damage caused by this procedure. Small changes in glucose levels were also observed in lambs that underwent tail docking with rubber rings, with a modest decrease observed at T_4_ and T_5_, likely as a result of animal stress and a slight decrease in energy reserves.

Electrophoretic data were subsequently performed, and they suggested the presence of an inflammatory status in lambs with tail docking by rubber rings, with changes in the albumin-to-globulin ratio [26]: at 7 days post-treatment, a drop in the proportion of albumin was observed in group B, in parallel to an increased proportion of α2-globulin and β-globulin. Tail docking by the surgical method did not significantly affect levels of diverse serum protein, with no significant changes in albumin and α2-globulin levels, and only a modest rise in β-globulin at T_2_ and T_4_, without statistical significance. In agreement, circulating levels of the other inflammatory marker transferrin revealed that tail forking by rubber rings, but not the surgical method, resulted in inflammation, with increased levels of transferrin at T_4_ (day 14). In the future, analysis of serum levels of key pro-inflammatory and anti-inflammatory cytokines should also be performed to provide a complete picture of the inflammatory status of sheep undergoing tail docking with these methods.

Finally, wool cortisol levels were quantified. Cortisol is the most commonly used physiological measure to assess the pain response caused by tail docking in sheep [3,9]. Previous studies reported that cortisol concentrations increased in lambs cauterised by both surgical and rubber ring methods [3,9,14,18,19]. On the other hand, the same sampling procedure (handling) might interfere with blood cortisol levels and can act as a confounder in the evaluation of any elevation in cortisol levels following tail docking [27,28,29]. In our study, cortisol levels were quantified in wool samples, considering that wool cortisol is regarded as a better predictor of stress than blood cortisol in sheep [30,31]. We observed that the control group exhibited higher wool cortisol values compared to the other groups, but with high inter-animal variability. After treatment (T_4_), group B (rubber ring) alone demonstrated a statistically significant increase in wool cortisol levels compared to pre-treatment levels (T_0_). These data suggest that stress was more intense in group B compared to the other groups, which is in line with serum levels of inflammatory markers (transferrin) and electrophoretic data.

## 5. Conclusions

Overall, our data suggest that the animals that underwent caudectomy with rubber rings presented inflammation and muscular damage related to the procedure and were stressed (with increased wool cortisol). Lambs that underwent tail docking with surgical treatment did not show an increase in inflammatory biomarkers or CPK serum levels, and no increase in wool cholesterol levels was detected.

The results obtained confirm that caudectomy in lambs, particularly that carried out by rubber rings, has a detrimental impact on animal welfare and results in stress and inflammation. These results are in agreement with the EFSA [32] and the Animal Welfare Indicators (AWIN) for sheep [33], where tail docking is regarded as a painful husbandry procedure, with a negative impact on sheep welfare [32,33]. This painful procedure is often performed without considering the negative impact on animals’ welfare [1]. Thus, the related disadvantages and benefits should be better re-evaluated. In addition, if caudectomy would be necessary, it should be performed by surgical procedures, which had less impact on the physiological parameters evaluated in this study.

## Figures and Tables

**Figure 1 animals-15-02614-f001:**
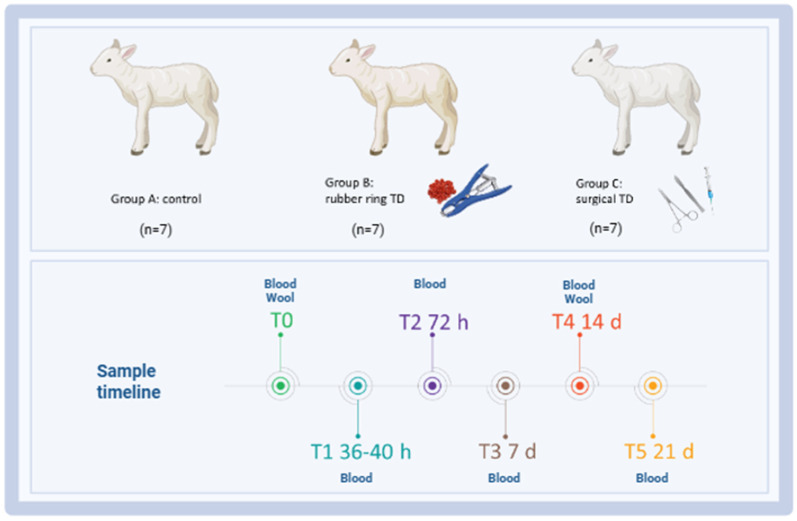
Schematic outline of the experimental study highlighting the key time points. Twenty-one lambs were enrolled in the study and randomly divided into three groups: group A (control group, *n* = 7), group B (tail docking by rubber ring, *n* = 7), and group C (tail docking by surgical amputation, *n* = 7). Serum samples were collected before treatment (T_0_) and five times after caudectomy: T_1_ (36–40 h), T_2_ (3 days), T_3_ (7 days), T_4_ (14 days), and T_5_ (21 days) for biochemical analysis and evaluation of inflammatory markers. At T_0_ and T_4_ (14 days), wool was also collected for cortisol evaluation. Image created with Biorender.com (accessed on 7 July 2025).

**Figure 2 animals-15-02614-f002:**
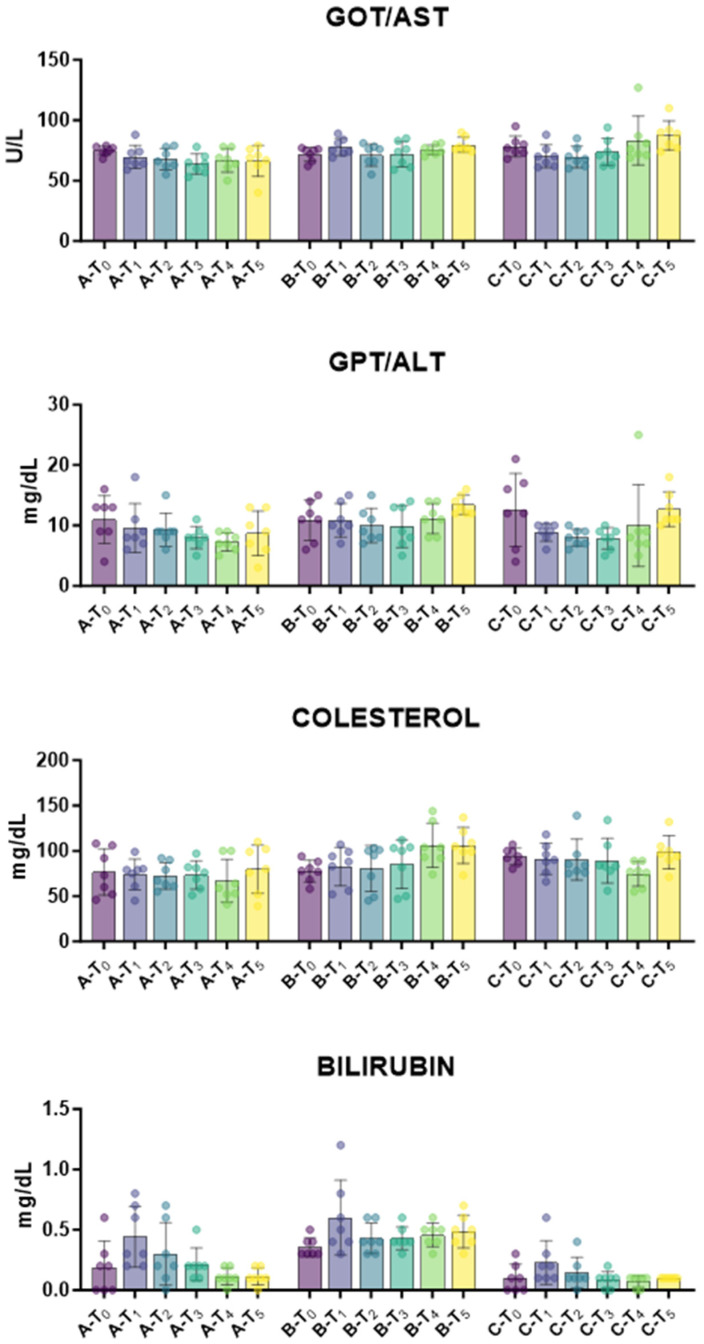
Kinetics of markers of hepatic function in serum samples taken throughout the experiment. Twenty-one lambs were enrolled in the study and randomly divided into three groups: group A (control group, *n* = 7), group B (tail docking by rubber ring, *n* = 7), and group C (tail docking by surgical amputation, *n* = 7). Serum samples were obtained through the experiment, and changes in the levels of GOT/AST, GPT/ALT, bilirubin, and cholesterol were monitored using an automated spectrophotometer. For each parameter, data post-treatment (T_1_, T_2_, T_3_, T_4_, and T_5_) were compared to the data pre-treatment (T_0_) using the parametric ANOVA followed by Dunnett’s multiple comparison tests or the non-parametric Kruskal–Wallis test followed by Dunn’s multiple comparison test.

**Figure 3 animals-15-02614-f003:**
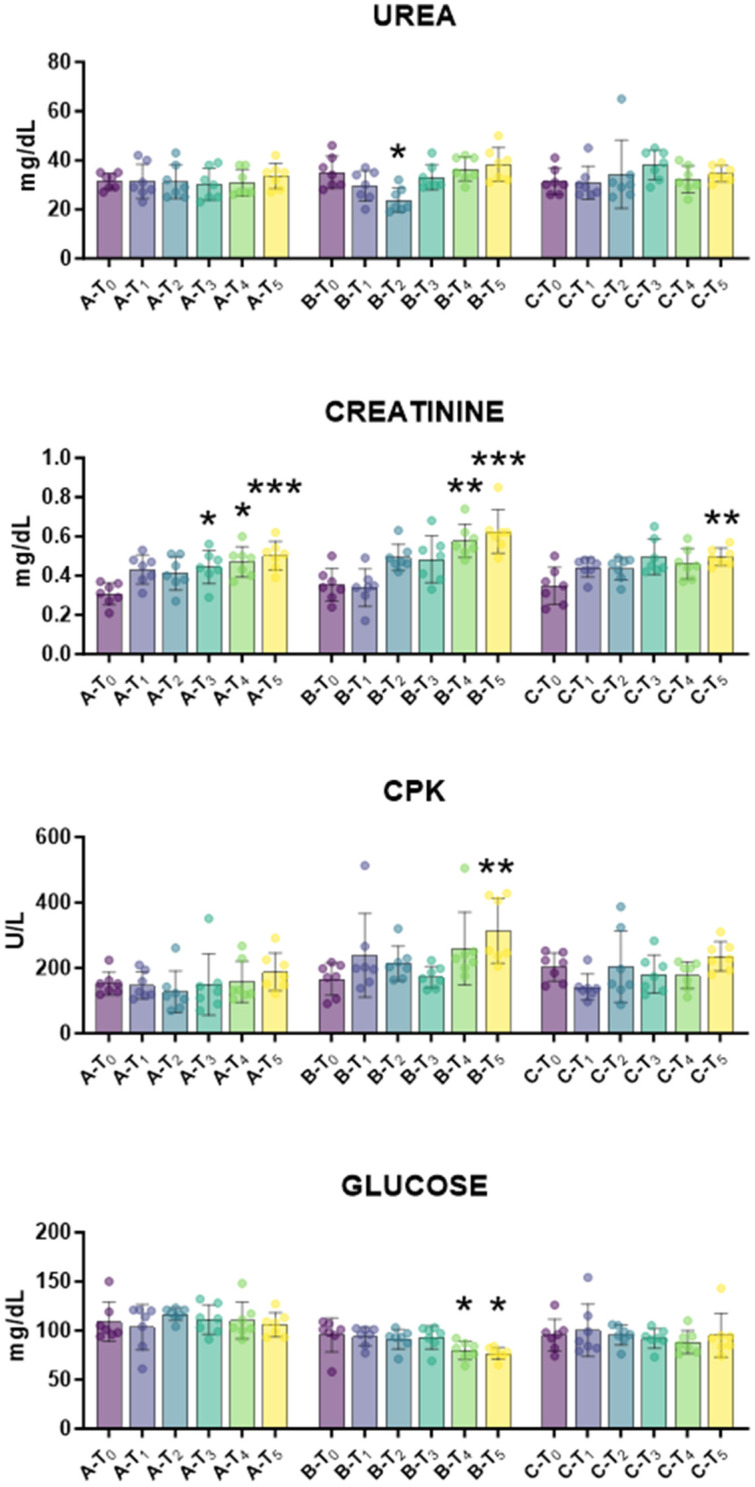
Kinetics of markers of renal function, CPK and glucose in serum samples taken throughout the experiment. Twenty-one lambs were enrolled in the study and randomly divided into three groups: group A (control group, *n* = 7), group B (tail docking by rubber ring, *n* = 7), group C (tail docking by surgical amputation, *n* = 7). Serum samples were obtained through the experiment, and changes in the levels of creatinine, urea, CPK, glucose were monitored using an automated spectrophotometer. For each parameter, data post-treatment (T_1_, T_2_, T_3_, T_4_, T_5_) were compared to the pre-treatment (T_0_) data using the parametric ANOVA followed by Dunnett’s multiple comparison tests or the non-parametric Kruskal–Wallis test followed by Dunn’s multiple comparison test; * *p* < 0.05, ** *p* < 0.01, *** *p* < 0.001.

**Figure 4 animals-15-02614-f004:**
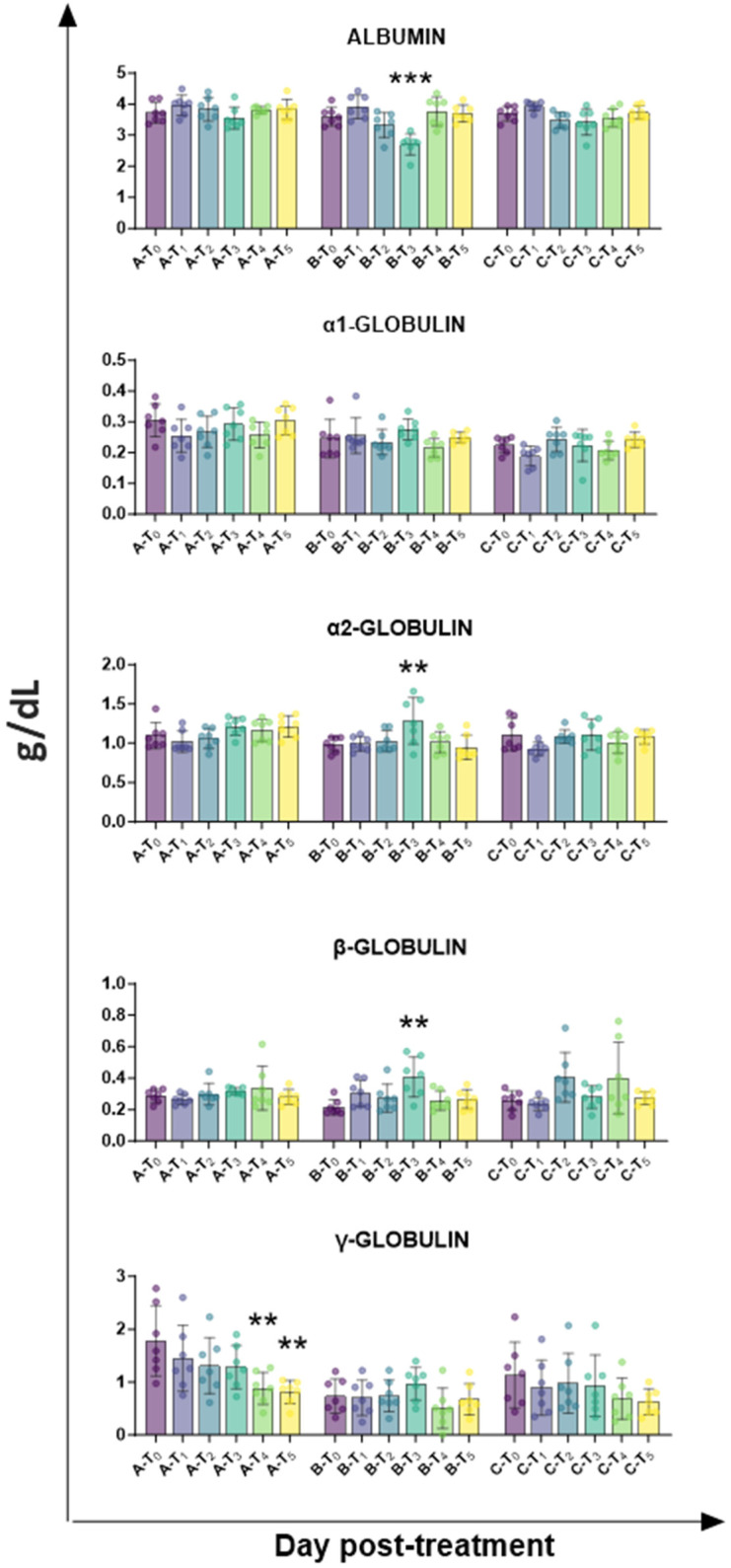
Kinetics of protein levels in serum samples taken throughout the experiment. Twenty-one lambs were enrolled in the study and randomly divided into three groups: group A (control group, *n* = 7), group B (tail docking by rubber ring, *n* = 7), and group C (tail docking by surgical amputation, *n* = 7). Serum samples were obtained through the experiment, and changes in the levels of total proteins were monitored using an automated spectrophotometer. Proportions of serum proteins (albumin, α1-globulin, α2-globulin, β-globulin, γ-globulin) were determined through electrophoretic analysis. For each parameter, data post-treatment (T_1_, T_2_, T_3_, T_4_, T_5_) were compared to the pre-treatment (T_0_) data using the parametric ANOVA followed by Dunnett’s multiple comparison tests or the non-parametric Kruskal–Wallis test followed by Dunn’s multiple comparison test; ** *p* < 0.01, *** *p* < 0.001.

**Figure 5 animals-15-02614-f005:**
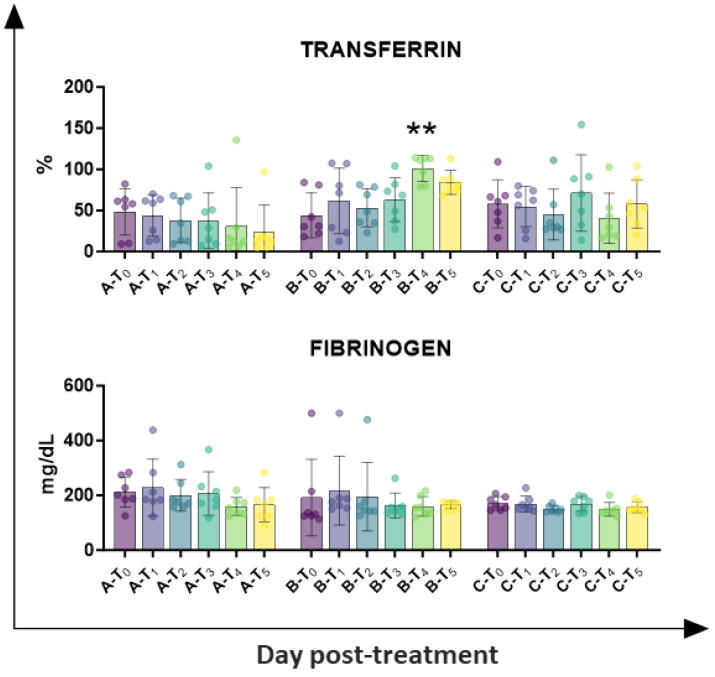
Kinetics of inflammatory markers in serum samples taken throughout the experiment. Twenty-one lambs were enrolled in the study and randomly divided into three groups: group A (control group, *n* = 7), group B (tail docking by rubber ring, *n* = 7), and group C (tail docking by surgical amputation, *n* = 7). Serum samples were obtained through the experiment, and changes in the levels of fibrinogen and transferrin were monitored using an automated blood coagulation analyser and a spectrophotometer, respectively. For each parameter, data post-treatment (T_1_, T_2_, T_3_, T_4_, T_5_) were compared to the pre-treatment (T_0_) data using the non-parametric Kruskal–Wallis test followed by Dunn’s multiple comparison test; ** *p* < 0.01.

**Figure 6 animals-15-02614-f006:**
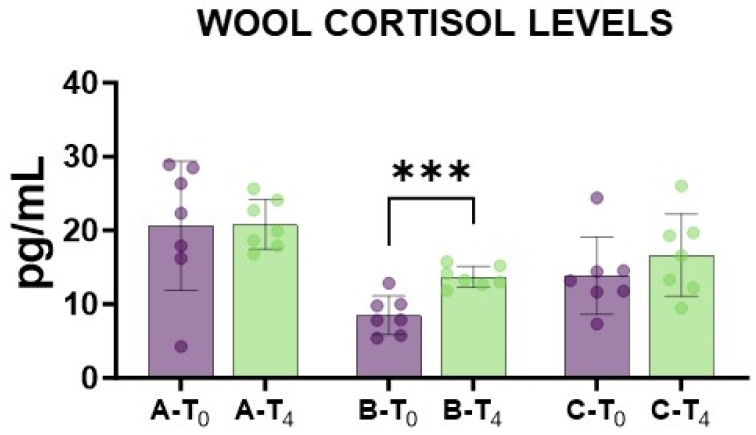
Impact of caudectomy on wool cortisol levels. Twenty lambs were enrolled in the study and randomly divided into three groups: group A (control group, *n* = 7), group B (tail docking by rubber ring, *n* = 7), and group C (tail docking by surgical amputation, *n* = 7). Wool samples were taken before treatment (T_0_) and at day 14 (T_4_), and their cortisol levels were determined. For each group, differences between the two time points were compared using an unpaired *t*-test; *** *p* < 0.001.

**Table 1 animals-15-02614-t001:** List of biochemical and electrophoretic parameters evaluated during the study, indicated as abbreviation, unit of measure, and by category.

Parameters (Acronym)	Unit of Measure	Exam Type
Total bilirubin	[mg/dL]	Metabolic profile
Creatinine	[mg/dL]	Metabolic profile
Total proteins	[g/dL]	Metabolic profile
Alanine aminotransferase (ALT/GPT)	[U/L]	Metabolic profile
Aspartateaminotransferase (AST/GOT)	[U/L]	Metabolic profile
Blood urea nitrogen (UREA)	[mg/dL]	Metabolic profile
Cholesterol	[mg/dL]	Metabolic profile
Creatine phosphokinase (CPK)	[U/L]	Metabolic profile
Glucose	[mg/dL]	Metabolic profile
Albumin	[%]	Electrophoresis
ALFA1 globulins	[%]	Electrophoresis
ALFA2 globulins	[%]	Electrophoresis
BETA globulins	[%]	Electrophoresis
GAMMA globulins	[%]	Electrophoresis
Transferritin	[%]	Metabolic profile
Fibrinogen	[mg/dL]	Coagulation profile

Unit of measures: %, percentage; g/dL, grams per decilitre; mg/dL, milligram per decilitre; U/L, units per litre.

## Data Availability

The raw data supporting the conclusions of this article will be made available by the authors on request.

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
