# Peer review of "Dynamics of Biochemical Parameters, Inflammatory and Stress Markers in Lambs Undergoing Caudectomy Using Two Different Methods"

_animals, 2025, doi:10.3390/ani15172614_

Round 1

Reviewer 1 Report

Comments and Suggestions for Authors

Overall, this manuscript is well-written, and the experimental design is adequate. The main problem is its lack of innovation; the effects of tail docking, while intuitive, have been extensively described many years ago. Indeed, most of the literature cited is more than 20 years old. 

Furthermore, the parameters the authors chose to evaluate are rather generic and dated, and, based on the statistically significant results obtained, it seems truly exaggerated to speak of an inflammatory state or muscle damage with a mild and transient increase in CK and transferrin. The authors could have considered more specific parameters, such as pro- and anti-inflammatory cytokines, to truly contribute to knowledge on the topic. Assessment of oxidative status could also have been a valid option.

Regarding stress assessment, some recent studies have demonstrated that blood, when sampled at the optimal time and method, remains the easiest and most accurate substrate for determining cortisol, compared to other long and time-expensive techniques such as wool. In this case, a comparison between blood and wool would have been interesting. Also, dealing with stress, a behavioral assessment would have been also interesting.

Before the paper can be considered for publication, the authors could consider to perform further measurements that could increase the scientific content of the work, further, the authors should completely revise the manuscript and limit themselves to highlighting, probably changing it into a short communication, the most innovative results, which in my opinion are: 1. the social and historical reasons behind the use of non-surgical techniques in some areas (which ones?), 2. the advantages of using wool for measuring cortisol (if any).

Reviewer 2 Report

Comments and Suggestions for Authors

Authors present an interesting idea and adequate study design.

Overall it is well designed and presented, but some additions are necessary in the animal method’s section. In studies like this, were pain is expected and behavior monitoring of the animals and clinical evaluation for any sigh of inappetence or illness are neccessary. These details  should be incorporated in the text. For example lambs were daily supervised daily and a veterinarian performed daily clinical evaluation for any sighs of behavior change. or illness. A lamb with pain changes its behavior before any change in blood biomarkers. Also details about feeding are necessary, for example the daily milk feeding protocol (suckling or fed with machine) is also useful. A lamb with any pain or problem after tail docking would have changes in milk consumption or suckling performance.

Discussion part is suggested to be more extended with more explaining on results. For example, the cortisol levels in control group are higher, almost double than the tail-cut groups, both at the beginning and at the second measurement. That is indeed an interesting result that deserves more discussion.

Some minor spelling mistakes in language:

L232, 239, 311, 332, 334 docking instead of dorking

L240 was

L296-in figure 6 title levels

Reviewer 3 Report

Comments and Suggestions for Authors

The manuscript presents a well-structured and ethically approved experimental study evaluating the impact of two caudectomy techniques—rubber rings and surgical amputation—on lamb welfare using a variety of physiological markers, including stress hormone (cortisol) levels and inflammatory and biochemical parameters. The work addresses an important and controversial animal welfare topic, providing novel data that could inform future guidelines and policy changes.
While the experimental design is sound and the analysis is largely appropriate, the manuscript has some areas that require improvement before publication. Key concerns include clarity in writing, typographical and grammatical errors, terminology inconsistencies (e.g., "dorking" vs. "docking"), and insufficient statistical discussion in the results.
1. Spelling Error in Title: “dorking” should be corrected to “docking” throughout the manuscript, including the title.
2. Abstract Clarity: Phrases such as “tail dorking” and “moderate raise” should be corrected for grammar and clarity (e.g., “moderate increase”).
3. Summary Accuracy: The abstract should better reflect the statistical significance and main findings of the study.
4. Introduction: Better structure the justification for comparing surgical vs. rubber ring caudectomy. Present benefits and concerns more concisely.
5. Literature Gaps: Clarify what existing studies have not covered and how this study uniquely addresses those gaps.
6. Creatinine Increase: The authors note creatinine increased in all groups—could this reflect age-related changes rather than treatment effects? Discuss further.
7. Overstatement Caution: Claims such as “this procedure results in pain and stress” should be tempered with discussion of sample size limitations and behavioral observations (which were not included).
8. Discuss how these findings align or contrast with international welfare standards or guidelines (e.g., OIE, EFSA).

Round 2

Reviewer 1 Report

Comments and Suggestions for Authors

The authors only partially addressed my concerns, but it's enough to make the manuscript acceptable for publication. Indeed, a limitation paragraph should be added highlighting that other, more accurate parameters may be investigated in the future to confirm their results.
